# Mirvetuximab soravtansine plus pembrolizumab in recurrent folate receptor alpha-positive uterine serous carcinoma: a phase II trial

Immune checkpoint inhibitors (ICI) synergize preclinically with antibody drug conjugates (ADC), harboring anti-tubulin maytansinoid payloads. We conducted an investigator-initiated, single-arm, phase 2 trial of mirvetuximab soravtansine (MIRV), a folate receptor alpha (FOLR1/FRα)-targeting ADC with the maytansinoid payload, DM4, combined with pembrolizumab in female patients with recurrent FOLR1-expressing serous endometrial cancer (EC, NCT03835819). Co-primary objectives include objective response rate (ORR) and rate of progression-free survival at 6 months (PFS6); secondary objectives include PFS, overall survival, duration of response and safety. Exploratory objectives include correlation of tumor genomics and immunoprofiling with clinical activity. Eighteen patients initiated protocol therapy [MIRV 6 mg/kg adjusted ideal body weight IV and pembrolizumab 200 mg IV every 3 weeks]. Confirmed ORR is 28% (1 complete and 4 partial responses, 95% CI:10-53%), Kaplan Meier estimate of PFS6 is 24.4% (95% CI:7.7-46.1%) with 4 patients progression free at 6 months; trial was closed early for feasibility (planned sample size of 35 patients not reached) and hence these results are considered preliminary. G3 treatment-related adverse effects were rare with no grade ≥4 toxicities. We report a population of high FOLR1-expressing tumor-associated macrophages (CD163 + FOLR1 +), suggesting potential on-target, off-tumor immune editing by MIRV. A composite biomarker score derived in this cohort correlates with objective response to MIRV and pembrolizumab.

Endometrial cancer (EC) is the most common gynecologic malignancy in the United States and has a rising incidence and mortality[1]. Approximately one third of EC patients present with locally advanced or metastatic disease, for which the five-year survival rates remain poor at 69% and <20%, respectively[2]. Immune checkpoint inhibitors (ICI) have demonstrated activity in advanced EC, both for recurrent disease[3-6] and in combination with chemotherapy and continued as maintenance in the first line setting[7-12]. However, ICI are less effective in mismatch repair proficient (pMMR) recurrent EC, with single agent ICI exhibiting response rates ranging from 3% to 14.1% in the pMMR cohorts across several clinical trials[4-6,13], highlighting an urgent need for alternative treatment regimens for recurrent EC.

ADCs represent a promising class of therapies showing impressive activity in numerous tumor types, including in EC[14]. At present, trastuzumab deruxtecan remains the only ADC approved for clinical use in recurrent HER2-expressing EC patients, with multiple other ADCs in development[14]. Mirvetuximab soravtansine (MIRV) is an ADC comprised of a folate receptor alpha (FOLR1/FRα)-targeted monoclonal

✉ e-mail: rebecca_porter@dfci.harvard.edu

antibody and the anti-tubulin maytansinoid payload, DM4, joined by a cleavable linker with a drug to antibody ratio of 3.5:1[15]. MIRV received Food and Drug Administration approval for the treatment of adult patients with FOLR1-positive, platinum resistant epithelial ovarian, fallopian tube and primary peritoneal cancers[16] based on the initial favorable ORR[17] seen in the phase 2 SORAYA trial, as well as a superior progression free (PFS) and overall survival (OS) compared with chemotherapy in the confirmatory phase 3 MIRASOL trial[18]. In PROC, high FOLR1 expression is defined via the PS2+ scoring method using the VENTANA FOLR1 RxDx assay[19], whereby ≥75% of tumor cells must display at least 2+ staining intensity by immunohistochemical (IHC) analysis. FOLR1 is also overexpressed and associated with poor prognosis in EC[20,21], suggesting its potential as a therapeutic target in this disease. In a Phase 1 dose expansion study of MIRV monotherapy in patients with FOLR1-positive advanced/recurrent EC (NCT01609556), the highest FOLR1 expression was reported in tumors of serous histology, and objective responses were observed in 2 of 24 (8%) patients and 11 patients (46%) achieved stable disease[22]. MIRV monotherapy was well-tolerated, potentially lending itself well to combinatorial strategies aimed at improving the modest single agent activity in EC.

ADCs can modulate the immune microenvironment through multiple mechanisms, including induction of immunogenic cell death, activation and maturation of dendritic cells and recruitment of other immune cells to the TME, stimulating both innate and adaptive immunity[23–28]. The combination of ADC and ICI have demonstrated synergism in preclinical studies[27,29] and promising anti-tumor activity in patients[30–33]. Moreover, in addition to target-mediated cytotoxicity, preclinical data has shown that MIRV activates monocytes and promotes phagocytosis of tumor cells through Fc-FcγR interactions[34]. Additionally, anti-tubulin payloads like DM4 promote DC maturation and migration to the draining lymph nodes, where they can activate naïve T cells[35,36]. Therefore, we hypothesized that MIRV combined with anti-PD-1 therapy would improve the response to ICI in patients with recurrent pMMR serous EC.

Here, we report the results of an investigator-initiated, two-stage, phase 2, single cohort study evaluating the activity of MIRV in combination with pembrolizumab in patients with recurrent/persistent EC (NCT03835819).

## Results

### FOLR1 expression in prescreening population
In the current study, a total of 144 eligible participants signed consent for pre-screening of archival tumor tissue for FOLR1 expression by central IHC using the anti-FOLR1 2.1 antibody developed by Ventana Medical Systems and following the PS2+ scoring methodology of quantifying the percentage of tumor cells staining at ≥2+ intensity. As shown in Table 1, of 137 patients with evaluable FOLR1 IHC results, 45 (32.8%), met the positivity cut off for trial eligibility with ≥50% of cells exhibiting at least 2+ membrane staining intensity (PS2+). The median percentage of cells per tumor sample expressing PS2+ staining was 30.0% (range: 0–100). Notably, 20 (14.6%) tumors were found to have PS2+ staining in ≥75% of tumor cells, the cut off for FOLR1-high expression selected in high grade serous ovarian cancer.

### Study population
Between December 30, 2019, and March 15, 2024, 18 patients were enrolled to the main study and received at least 1 dose of MIRV and pembrolizumab (Fig. 1). All patients were female and had histologically confirmed uterine serous adenocarcinoma with intact MMR protein expression by IHC. Additional baseline patient characteristics are summarized in Table 2; 8 patients (44.4%) had received prior ICI therapy, whereas 10 patients (55.6%) were ICI naïve. The median FOLR1 expression based on percentage of cells with at least PS2+ expression was 72.5% (range: 50.0, 98.0); 50% of enrolled patients had FOLR1 ≥ 75%. At the data cutoff of July 05, 2024, the median follow-up for OS

was 11.3 months (IQR: 3.3-NR months). No patients were remaining on study.

### Antitumor activity of mirvetuximab and pembrolizumab
A total of 18 patients were evaluable for efficacy analyses; one patient withdrew from the study for non-AE reasons after receiving 3 doses of study treatment. Among the first 5 patients enrolled, we observed 2 confirmed objective responses meeting the criteria to continue enrollment in the two-stage design. Among all 18 patients, the confirmed ORR was 28% (95% CI: 10-53%) including 1 confirmed complete response and 4 confirmed partial responses (Fig. 2A). While the study meets the required number of responses for ORR (≥ 4 objective responses) specified by the two-stage design, and the 95% confidence interval excludes the null ORR of 5%, we did not complete enrollment of the trial to the planned sample size of 35 patients and hence these results should be considered preliminary. There were also 2 unconfirmed partial responses. An additional 4 patients had stable disease of any duration as best response. Depth and duration of response by subject are shown in Fig. 2B and Supplementary Fig. 1, respectively. FOLR1 expression and prior history of ICI (anti-PD-1 or anti-PD-L1) therapy is noted for each patient. There were 4 patients alive and progression-free at 6 months, resulting in a PFS6 rate of 22% (95% CI: 6–48%).

The observed median PFS (mPFS) was 2.73 months (95% CI: 1.2–4.5 months) (Supplementary Fig. 2) and the PFS at 6 months was 24.4% (95% CI 7.7–46.1%) by the Kaplan–Meier method. Median OS for the overall cohort was 12.5 months (95% CI: 5.2-NR months). The observed median duration of response was 8.17 months (range: 2.76–18.14 months) in the 5 patients with confirmed responses. Notably, in the patient with a complete response, despite discontinuation of pembrolizumab at 2.8 months from treatment initiation for adrenal insufficiency, she exhibited a sustained response of 18.4 months before discontinuing mirvetuximab for pneumonitis. Another patient achieved a deep partial response (50.5% tumor reduction as best response) that was sustained for at least 10.1 months, including on last assessment. Mirvetuximab and pembrolizumab were both discontinued in that patient at 12.2 months from treatment initiation for toxicities.

The initial design of the trial enrolled ICI naïve patients only, but a subsequent protocol amendment allowed for enrollment of up to 19 patients with prior ICI targeting the PD-1/PD-L1 pathway. In an exploratory subgroup analysis, we evaluated the efficacy of mirvetuximab and pembrolizumab in patients that were ICI naïve (n = 10) vs exposed (n = 8). All 5 confirmed responses were in patients who were ICI naïve; 2 patients with prior ICI therapy exhibited unconfirmed responses. Among the 11 patients with either a response or stable disease, 8 were ICI naïve while 3 had received prior ICI therapy (Supplementary Table 1). In contrast, 4 of 6 patients with progressive disease as best response had received prior ICI. All 4 patients who achieved PFS6 were ICI naïve. The mPFS for the ICI-naïve and ICI-exposed groups were 4.5 months (95% CI: 1.15, NR) and 1.2 months (95% CI: 1.12, NR), respectively (Supplementary Fig. 3A). Prior ICI exposure was significantly associated with shorter PFS (HR = 0.16; 95% CI 0.04−0.63; p = 0.003) and a significantly lower response rate (Fisher's exact test p-value = 0.036, ORR is 0% [95% CI: 0, 37%] in the ICI-exposed cohort, and 50% [95% CI: 19, 81%] in the ICI-naïve cohort), however these results are all preliminary given small sample sizes in the subgroups of interest.

### Safety
Safety and tolerability were as expected for this two-drug combination. Of the 18 patients who initiated treatment with this regimen, 18 (100%) reported a treatment-related adverse event (TRAE) of any grade (Supplementary Table 2). The most common TRAEs of any grade occurring in >25% of patients were diarrhea (n = 8, 44%), fatigue (n = 8,

44%), AST elevation ($n = 8$, 44%), blurred vision ($n = 7$, 39%), anemia ($n = 6$, 34%) and peripheral sensory neuropathy, decreased platelet count, and anorexia ($n = 8$, 28% for each). Grade 3 TRAEs were rare and are presented in Table 3; these included diarrhea, anemia, thrombocytopenia, neutropenia, peripheral sensory neuropathy, hypertension, corneal ulcer and adrenal insufficiency, each occurring in 1 participant (6%). There were no grade 4 or 5 toxicities reported.

A total of 4 (22.2%) and 4 (22.2%) patients discontinued mirvetuximab and pembrolizumab, respectively, owing to treatment-related toxicities. Among patients who discontinued study drugs for toxicities, the median number of completed cycles was 10 cycles (IQR:6.5–15.5) for mirvetuximab and 6.5 cycles (IQR: 5.5–8.5) for pembrolizumab. Dose delays and/or reductions were permitted for mirvetuximab according to the investigator brochure[37]; 8 patients (44.4%) required a dose delay, and 7 patients (38.9%) required dose reductions.

Based on the known safety profile of mirvetuximab, ocular events were expected and were managed by prophylactic mitigation strategies, including lubricating and steroid eye drop use, and through dose modifications as needed per protocol. Additionally, ocular surface adverse events, including rare cases of corneal ulcer, have been reported in association with ICIs, including pembrolizumab[38,39]. Ocular adverse events occurred in 10 participants (55.6%); the most common were blurred vision (39%), keratitis (11%), and dry eyes (11%). All ocular adverse events were grade 1 or 2 except for one participant (6%) with a grade 3 corneal ulcer. Immune-related AEs occurred as expected based on the known safety profile of pembrolizumab. There were two participants (11%) with grade 2 pneumonitis and one participant each (6%) with grade 3 adrenal insufficiency and grade 2 hyperthyroidism.

## Biomarker analyses

Given the encouraging ORR and the prolonged duration of response in some patients, we aimed to identify potential biomarkers of response to MIRV and pembrolizumab in this cohort of patients. In post-hoc exploratory analyses, we first examined FOLR1 expression levels and performed genomic and immune profiling of archival tumor tissue and assessed whether any biomarkers were individually associated with objective response or PFS. With regard to FOLR1 expression, 4 of the 5 confirmed responses occurred in patients with FOLR1 expression of 75% or higher; one patient with FOLR1 expression <75% achieved a confirmed PR. The confirmed ORR in patients with FOLR1 expression 75–100% was 44% (95% CI: 14%, 79%), whereas the confirmed ORR was 11% (95% CI: 0%, 48%) in patients with FOLR1 expression between 50% and 74% (Supplementary Table 3). However, there was not a statistically significant association between FOLR1 expression level and objective response ($p = 0.29$). The mPFS for FOLR1-moderate and FOLR1-high tumors was similar at 2.63 (95% CI: 1.12, 4.50) and 2.73 (95% CI 1.18, 8.25) months, respectively (Supplementary Fig. 3B).

Tumor genomic profiling and IHC analyses (ER, HER2) were either previously available or performed on archival FFPE specimens from 17 patients in the study. All 17 (100%) tumors were MMR proficient (pMMR) or microsatellite stable (MSS) as assessed by IHC or PCR on next generation sequencing (NGS) and all harbored *TP53* mutations; no tumors were polymerase epsilon (*POLE*) mutated. Table 4 presents the associations between specific genomic and protein biomarkers and objective response; all missing cases were excluded when calculating the *p*-value. There were 3 tumors with *CCNE1* amplifications, and all of these were in patients who had an objective response ($p = 0.02$). There were no other statistically significant associations between HER2 and ER protein expression by IHC, tumor mutational burden (TMB), or specific genomic alterations, including *MYC and ERBB2* amplifications, with clinical activity.

We next performed immune profiling of archival tumors to identify alternative biomarkers of clinical activity to this ADC and ICI combination. Immune profiling of 17 tumors was performed on archival FFPE specimens using multiplexed immunofluorescence (ImmunoProfile). Staining with two separate panels of six markers each

## Table 1 | FOLR1 expression in prescreening cohort

|  | All Prescreened Tumors $N = 137$ |
|---|---|
| **PS2+ value** |  |
| Mean (SD) | 33.9 (29.9) |
| Median [Min, Max] | 30.0 [0, 100] |
| <25% | 62 (45.3%) |
| 25–49% | 30 (21.9%) |
| 50–74% | 25 (18.2%) |
| ≥75% | 20 (14.6%) |

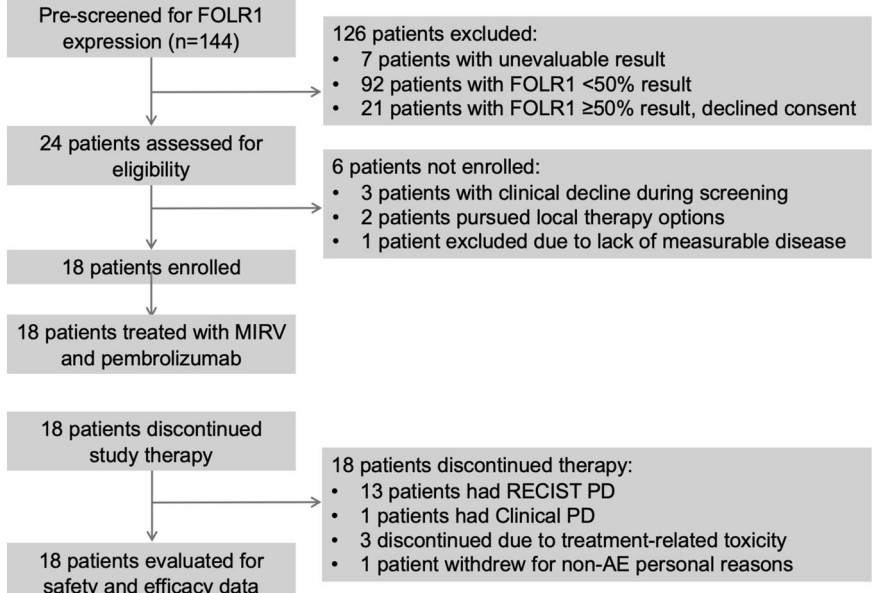

**Fig. 1** | Study overview.

(Supplementary Tables 4 and 5) was performed on tumor specimens, and the count density (positive cells per mm²) and percentage of positive cells was determined for each marker. All 17 tumors were stained for cytokeratin (CytoK, tumor cells), CD8 (CD8 + T cells), CD163 (macrophages), PD-L1, and FOLR1 (Fig. 3A), and 15 tumors were additionally stained for FOXP3 (regulatory T cells) and PD-1. Notably,

### Table 2 | Demographic and baseline characteristics of patients

|  | Recurrent pMMR/MSS serous EC (*N* = 18) |
|---|---|
| **Age (years)** | |
| Median (min, max) | 67.2 (62.7, 78.5) |
| **Ethnicity** | |
| Hispanic or Latino | 2 (11.1%) |
| Non-Hispanic | 16 (88.9%) |
| **Race** | |
| White | 15 (83.3%) |
| Black or African American | 1 (5.6%) |
| Asian | 1 (5.6%) |
| Other | 1 (5.6%) |
| **Stage at diagnosis** | |
| I | 5 (27.8%) |
| III | 7 (38.9%) |
| IV | 6 (33.3%) |
| **ECOG Performance Status** | |
| 0 | 12 (66.7%) |
| 1 | 6 (33.3%) |
| **Prior lines of therapy** | |
| 1 | 2 (11.1%) |
| 2 | 7 (38.9%) |
| 3 | 5 (27.8%) |
| 4 | 4 (22.2%) |
| **FOLR1 % PS2+ staining** | |
| Mean (SD) | 72.9 (14.4) |
| Median (min, max) | 72.5 (50.0, 98.0) |
| PS2 + 50–74% | 9 (50.0%) |
| PS2 + 75–100% | 9 (50.0%) |
| **Prior anti-PD-1/PD-L1** | |
| ICI-exposed | 8 (44.4%) |
| ICI-naïve | 10 (55.6%) |

we first compared FOLR1 IF to the FOLR1 IHC performed in pre-screening for trial eligibility. Although the two assays were performed on non-contiguous tumor sections, we observed an overall correlation between FOLR1 cell density by IF and FOLR1 PS2+ by IHC (*r* = 0.432; Supplementary Fig. 4). The immune profile of tumors was compared between patients with confirmed objective response vs no objective response (Table 5). While overall PD-L1 and FOLR1 cell densities in tumor sections were not independently significantly associated with objective response, the PD-L1 cell density in tumor cells (PD-L1 + CYTOK+; *p* = 0.035) and in FOLR1-expressing cells (PD-L1 + FOLR1+; *p* = 0.045) was significantly associated with objective response. In addition, we observed a striking presence of CD163 + FOLR1+ cells in 3 of 5 patients with objective responses (Fig. 3B), though cell density of this marker combination was not statistically associated with objective response. No additional statistically significant associations between any single biomarker or double positive population of interest (e.g., CD8 + PD-1+, CD163 + PD-L1) with clinical responses were detected.

Based on the observations of higher FOLR1 expression and markers of an inflamed tumor on immune profiling without strong individual associations with objective response or PFS to MIRV and pembrolizumab, we reasoned that a combination of these markers may be predictive of response. Therefore, we defined a composite Biomarker score (B score), comprised of high vs low scoring of FOLR1 IHC and CD8+, PD-L1+, and CD163 + FOLR1+ cell densities. The median B score was significantly higher in patients with an objective response (2; IQR: 2.00–2) compared with non-responders (1; IQR: 0.75–2; *p* = 0.015; Fig. 3C) and Biomarker positive status (B+), defined as tumors with 2 or more biomarkers in the High category, was positively correlated with objective response (two-sided Fisher exact test, *p* = 0.026; Fig. 3D). Patients with B+ tumors also demonstrated a non-significant trend toward improved PFS (*p* = 0.061; Supplementary Fig. 5). Median OS was 1.28 months (95% CI: 1.18–NR) for patients with B − scores (*n* = 7, 6 events) compared with 6.19 months (95% CI: 2.73–NR) for patients with B+ scores (*n* = 8, 5 events), supporting the potential clinical significance of this hypothesis-generating biomarker score, or others similarly describing a favorable tumor and immune microenvironment, in predicting patients who may derive benefit from an ADC combined with ICB.

## Discussion

In this investigator-initiated phase 2 study, the combination of MIRV and pembrolizumab demonstrated an ORR of 28% in patients with recurrent pMMR high-grade serous EC, including one patient with a complete response and two patients who achieved prolonged complete and partial responses of at least 19.6 and 10.1 months,

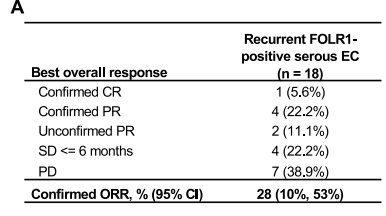

**A**

| Best overall response | Recurrent FOLR1-positive serous EC (n = 18) |
|---|---|
| Confirmed CR | 1 (5.6%) |
| Confirmed PR | 4 (22.2%) |
| Unconfirmed PR | 2 (11.1%) |
| SD <= 6 months | 4 (22.2%) |
| PD | 7 (38.9%) |
| **Confirmed ORR, % (95% CI)** | **28 (10%, 53%)** |

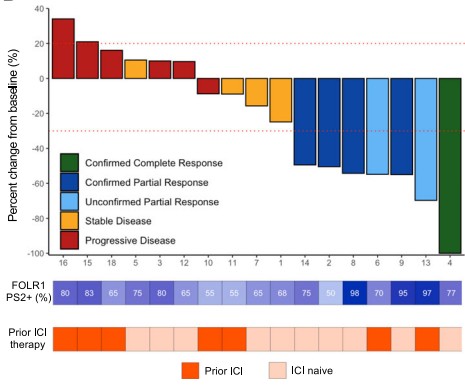

**B**

**Fig. 2 | Clinical responses to MIRV and pembrolizumab. A** Best responses in all patients evaluable for RECIST 1.1 response. **B** Waterfall plot of best percent change in aggregate size of target lesions per RECIST v1.1 criteria for all 18 patients. FOLR1 PS2+ expression and prior ICI status are included for each patient underneath.

(1) One patient was excluded from waterfall plot because of failure to reach first restaging scan. (2) No patients remain on study treatment. CR complete response, PD progressive disease, PR partial response, SD stable disease.

**Table 3 | Treatment-related adverse events of grade ≥ 3**

| Adverse event[a] | Maximum grade | | | |
|---|---|---|---|---|
| | Grade 1 | Grade 2 | Grade 3 | All Grades |
| Diarrhea | 4 (22%) | 3 (17%) | 1 (6%) | 8 (45%) |
| Anemia | 3 (17%) | 2 (11%) | 1 (6%) | 6 (34%) |
| Peripheral sensory neuropathy | 1 (6%) | 3 (17%) | 1 (6%) | 5 (29%) |
| Platelet count decreased | 2 (11%) | 2 (11%) | 1 (6%) | 5 (28%) |
| Neutrophil count decreased | 1 (6%) | 0 (0%) | 1 (6%) | 2 (12%) |
| Hypertension | 1 (6%) | 0 (0%) | 1 (6%) | 2 (12%) |
| Adrenal insufficiency | 0 (0%) | 0 (0%) | 1 (6%) | 1 (6%) |
| Corneal ulcer | 0 (0%) | 0 (0%) | 1 (6%) | 1 (6%) |
| Fall | 0 (0%) | 0 (0%) | 1 (6%) | 1 (6%) |
| Spinal fracture | 0 (0%) | 0 (0%) | 1 (6%) | 1 (6%) |
| Dizziness | 0 (0%) | 0 (0%) | 1 (6%) | 1 (6%) |
| Vascular disorders - Other | 0 (0%) | 0 (0%) | 1 (6%) | 1 (6%) |

[a]The denominator to all calculated percentages is 18, the number of patients who received at least one dose of study drug.

**Table 4 | Genomic biomarker associations with clinical activity**

| ALTERATION | Objective response | | |
|---|---|---|---|
| | Yes (N = 5) | No (N = 12) | p value[a] |
| **ER positive (≥ 1% by IHC)** | | | 1.00 |
| NO | 0 (0.0%) | 1 (100.0%) | |
| YES | 2 (28.6%) | 5 (71.4%) | |
| Missing | 3 | 6 | |
| **HER2 by IHC** | | | 0.54 |
| 0–1+ | 3 (50.0%) | 3 (50.0%) | |
| 2+–3+ | 1 (16.7%) | 5 (83.3%) | |
| Missing | 1 | 4 | |
| **Tumor Mutational Burden > 10 Mut/MB** | | | 1.00 |
| NO | 5 (33.3%) | 10 (66.7%) | |
| YES | 0 (0.0%) | 1 (100.0%) | |
| Missing | 0 | 1 | |
| **CCNE1 amplification** | | | 0.02 |
| NO | 2 (16.7%) | 10 (83.3%) | |
| YES | 3 (100.0%) | 0 (0.0%) | |
| Missing | 0 | 2 | |
| **MYC amplification** | | | 1.00 |
| NO | 4 (33.3%) | 8 (66.7%) | |
| YES | 1 (33.3%) | 2 (66.7%) | |
| Missing | 0 | 2 | |
| **ERBB2 amplification** | | | 0.60 |
| NO | 4 (40.0%) | 6 (60.0%) | |
| YES | 1 (20.0%) | 4 (80.0%) | |
| Missing | 0 | 2 | |

[a]All statistical tests were two-sided Fisher's Exact tests, and no adjustment was made for multiple comparisons.
Source data are provided as a Source Data file.

respectively, meeting prespecified criteria to be considered worthy of additional investigation. Moreover, this trial enrolled a population of patients with biologically aggressive disease; eligible patients were required to have predominantly serous histology tumors, all of which were *TP53*-mutated, and may have received up to 4 prior lines of therapy, including prior ICI. There were no new safety signals or unexpected toxicities with this combination, as has also been the case with other ADC combinations[31,40,41]. Thus, the promising signal of activity and the favorable safety profile in this report of combined ADC and ICI therapy in recurrent EC lend support to the potential efficacy of this approach.

This study aimed to address the critical need for novel therapies for recurrent pMMR EC, given that single agent ICI have modest efficacy in this setting[13] and the combination of pembrolizumab and lenvatinib has improved activity but with added toxicity[42]. Moreover, biomarkers predictive of response to pembrolizumab and lenvatinib have not been identified despite comprehensive translational studies including immune profiling and genomic analyses[43]. Here, the ORR of 28% to MIRV and pembrolizumab is favorable in this population, yet the wide confidence intervals and variable duration of response, with some responders exhibiting prolonged responses while others experienced disease progression shortly after response, underscores the importance of identifying predictive biomarkers and to elucidate mechanisms of resistance. To this end, we first examined the associations between FOLR1 expression and immune phenotypes with clinical activity. While there was a non-statistically significant trend toward improved response rates to MIRV and pembrolizumab in patients with higher FOLR1 PS2+ score, FOLR1 expression was not correlated with PFS. Further, the patient with a prolonged CR had the lowest FOLR1 PS2+ score, highlighting that FOLR1 expression alone does not reliably predict response. Likewise, all patients with confirmed responses were ICI naïve; however, no statistically significant association between prior ICI exposure and clinical response was noted in the overall population. Given the smaller size of this study, associations between clinical activity and FOLR1 expression and/or prior ICI exposure will need to be confirmed in future studies. This consideration is particularly relevant with the recent paradigm shift to incorporating ICI early in the treatment course of patients with advanced EC. Our findings further underscore the importance of utilizing biomarkers to guide

selection of optimal ICI combinations in patients with prior ICI exposure.

Given that FOLR1 expression and prior ICI status could not clearly predict clinical responses to MIRV and pembrolizumab, we examined the tumor immune microenvironment (TIME) through multiplex immunofluorescence, or ImmunoProfile, analysis. Moreover, 3 of the 5 patients with objective responses were found to have *CCNE1*-amplified tumors, which are enriched in uterine serous carcinomas[44] and has been associated in multiple tumor types with poor prognosis and relative chemoresistance[45–47]. Interestingly, more recent reports have reported increased tumor-infiltrating immune cells and response to immunotherapy in *CCNE1*-amplified tumors[48,49], though the association of *CCNE1* amplification and overexpression with anti-tumor immunity remains unclear in EC. In this patient cohort, we observed markers of tumor inflammation, including high density of CD8+ and PD-L1+ cells more frequently in patients with objective responses, though these were not individually predictive of response. To extend our search for biomarkers, we also included IF for FOLR1, initially to ask whether FOLR1+ tumor cells were also PD-L1+. This incorporation of FOLR1 into multiplexed immunofluorescence profiling facilitated the detection of a striking enrichment of FOLR1+ tumor-associated macrophages (CD163+) in some tumors, including in 3 of 5 patients with confirmed responses. Interestingly, while tumor-associated macrophages (TAMs) are known to express FOLR2[50,51], our observation of a significant FOLR1 + CD163+ population in some patients has not previously been described. Expression of FOLR1 on TAMs raises the

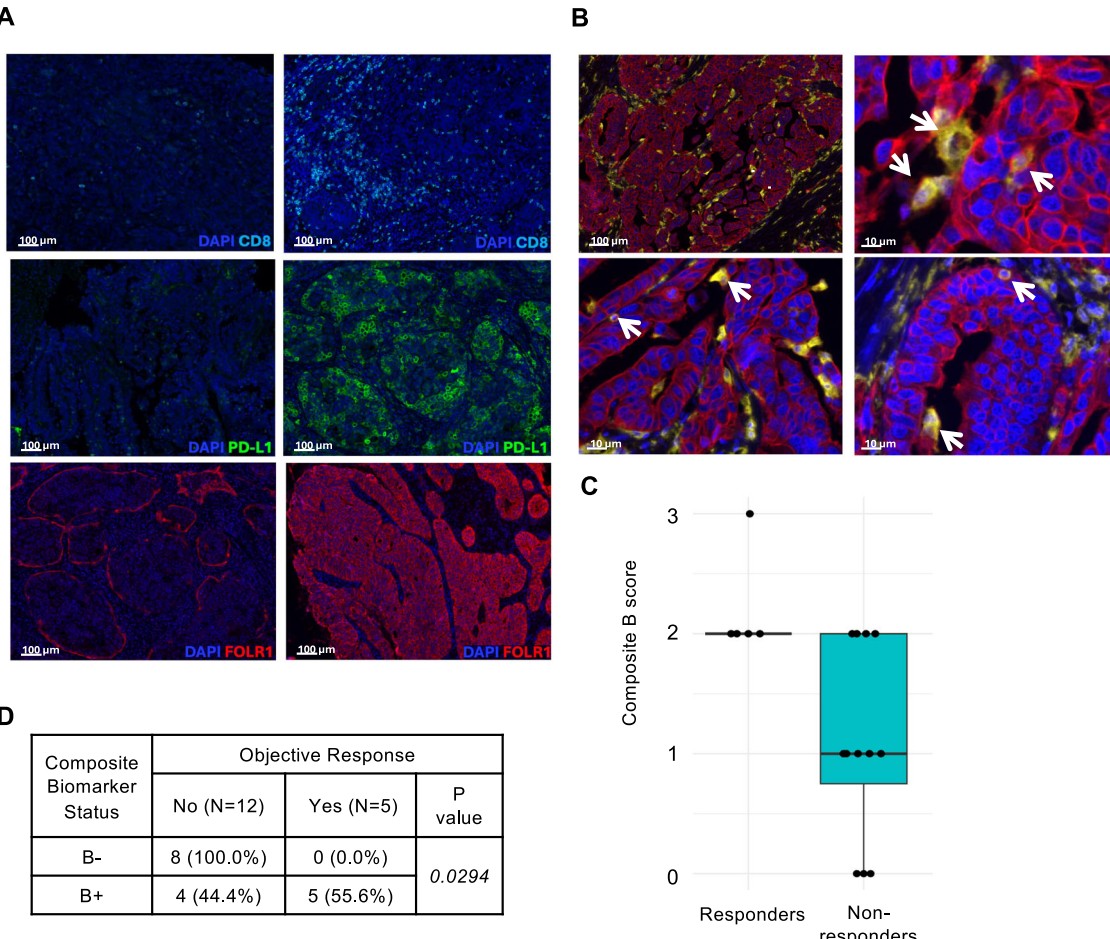

**Fig. 3 | Multiplex immunofluorescence (mIF) analyses reveal composite bio-marker (B+) score associated with objective response to MIRV and pem-brolizumab. A** Representative single marker images (one 20× field of view out of six total quantified per sample) showing CD8 (cyan), PD-L1 (green), FOLR1 (red), and DAPI (Blue) in tumor sections with low (left panels) and high (right panels) cell density for each marker. Selected images are representative of the 17 tumors with mIF. **B** Representative images highlighting cells (arrows) with double positive staining for CD163 (yellow) and FOLR1 (red) signifying a population of FOLR1+ CD163 + tumor-associated macrophages. Selected images are representa-tive of the 17 tumors with mIF. **C** Distribution of composite biomarker (B) scores by objective response status. Responders ($n = 5$) had significantly higher B scores compared with non-responders (NR, $n = 12$; median 2 [IQR 2–3] vs 1 [IQR 0.25–1.75], range 2–4 vs 0–2; Wilcoxon rank-sum test, $p = 0.01$). The B score was calculated by assigning 1 point for a high value in each of the following biomarkers: FOLR1 IHC, CD8+ cell density (positive cells per mm²), PD-L1+ cell density, and CD163+FOLR1+ cell density. **D** Correlation of (B) score status with objective response. *P*-value from Fisher's exact test is indicated. All statistical tests were two-sided, and no adjust-ment was made for multiple comparisons. Source data are provided as a Source Data file.

## Table 5 | ImmunoProfile results

| Median (IQR) | Objective response | | | |
| --- | --- | --- | --- | --- |
| | Yes (*N* = 5) | No (*N* = 12) | Total (*N* = 17) | *p* value[a] |
| **CD8+** | 289.3 (58.4, 361.4) | 65.6 (41.3, 130.4) | 74.4 (41.8, 261.2) | *0.34* |
| **CD163 + PDL1+** | 77.0 (63.6, 250.9) | 29.0 (16.1, 85.4) | 63.6 (22.3, 92.1) | *0.17* |
| **CD163+** | 409.9 (183.6, 555.6) | 335.7 (168.4, 458.2) | 380.8 (175.2, 555.6) | *0.53* |
| **CD163 + FOLR1+** | 88.0 (20.0, 186.2) | 68.4 (35.2, 112.9) | 81.8 (26.3, 148.4) | *0.75* |
| **CYTOK + FOLR1+** | 3824.7 (1755.4, 4593.4) | 3441.8 (2431.3, 4338.4) | 3557.0 (1935.5, 4593.4) | *1.00* |
| **CYTOK + PDL1+** | 302.7 (85.6, 329.7) | 77.5 (34.2, 114.2) | 85.6 (53.6, 184.4) | *0.04* |
| **FOLR1+** | 4541.9 (2984.2, 4972.6) | 4489.1 (3090.8, 4883.7) | 4541.9 (3012.3, 4926.6) | *0.92* |
| **PDL1+** | 707.8 (467.8, 1079.0) | 196.8 (109.4, 474.2) | 399.3 (113.0, 707.8) | *0.07* |
| **PDL1 + FOLR1+** | 118.8 (90.5, 428.0) | 71.8 (34.3, 112.5) | 89.3 (43.6, 120.2) | *0.05* |
| **CD8 + CD163+** | 5.3 (4.8, 6.8) | 3.0 (1.1, 9.3) | 4.8 (1.3, 8.8) | 0.53 |
| **CD8 + FOLR1+** | 25.1 (24.4, 36.9) | 11.7 (3.0, 23.0) | 16.3 (7.3, 25.1) | 0.11 |

[a]All statistical tests were two-sided Wilcoxon rank-sum tests, and no adjustment was made for multiple comparisons.
Source data are provided as a Source Data file.

possibility that ADC target expression on pro-tumorigenic immune cells in the TIME could enhance the activity of the ADC via on-target, off-tumor elimination of these cells. On treatment biopsies in future studies could be utilized to determine if immune cells expressing FOLR1 are diminished after treatment with MIRV. This potential on-tumor, off-target effect of MIRV on TAMs, if verified in future studies, could expand the pool of potential biomarkers for response to ADC and ICI therapies.

While we found that no single immune-related biomarker or FOLR1 expression could individually predict response to MIRV/pembrolizumab, we hypothesized that a combination of tumor and TIME characteristics may represent a predictive biomarker for response to this combination. We therefore selected 2 markers each with relevance to either MIRV or pembrolizumab, including FOLR1 PS2+ expression and FOLR1+/CD163+ cell density for MIRV, and CD8+ cell density and PD-L1 CPS for pembrolizumab. Indeed, a composite Biomarker (B) score comprised of these four markers derived in this cohort was associated with clinical response to MIRV and pembrolizumab, suggesting potential predictive ability of this B score for PFS. However, this score was derived and evaluated in this small cohort and should therefore be considered hypothesis-generating. Validation in external cohorts of patients in the future will be required to draw conclusions on the clinical relevance of this B score.

Notably, this study enrolled patients with moderate to high FOLR1 expression of at least 50% PS2+ by central IHC analysis, which required prescreening a large population to identify enough patients to meet enrollment. Numerous studies have reported high rates of FOLR1 overexpression in ovarian cancers, particularly in those of serous histology[52–56], yet relatively few studies have reported on FOLR1 expression patterns across ECs. In one, *FOLR1* mRNA was detected at 60x higher levels by quantitative PCR in tumors of serous histology compared with normal endometrial tissue[57]. In another study, *FOLR1* overexpression was reported in 65/411 (16%) of EC tumors on tissue microarray, with serous carcinoma (USC) tumors demonstrating the highest rate of overexpression at 48% (23/48)[20]. Across 4 studies examining FOLR1 protein expression in small numbers of EC cases, FOLR1 overexpression was reported in 31/46 cases[58–61]. Recently, IHC analysis of FOLR1 protein expression was examined in a cohort of 111 advanced EC patients from a single institution, with 18.9% of cases reported as FOLR1 high, defined in this study as ≥25% PS2 + FOLR1 staining[62]. Here, we report FOLR1 expression in 137 patients with pMMR serous EC. High FOLR1 expression, defined here as ≥50% PS2+ staining, was observed in 32.8% of tumors. This higher positivity rate in our study despite utilizing a higher PS2+ cutoff is likely due to the selection of serous histology tumors, which are characterized by a high frequency of *TP53* mutations[63]. Indeed, in the report by Xiao et al, high FOLR1 expression was seen most frequently in *TP53*-mutated tumors[62]. Together, these studies improve our understanding of FOLR1 expression across EC subsets and suggest that FOLR1-targeting strategies may be most beneficial in patients with *TP53*-abnormal tumors. Ongoing studies of FOLR1 expression across molecular subsets of EC will confirm any association of FOLR1 overexpression with *TP53* status or other molecular subsets.

In the current study, we utilized a FOLR1 PS2+ cutoff of 50% for eligibility. This was based on observations from a phase Ib study of MIRV combined with pembrolizumab in 14 patients with PROC, wherein 5 of the 6 confirmed responses were in patients with at least moderate FOLR1 (PS2+ ≥ 50%) expressing tumors, and only 1 of 6 patients with low FOLR1 expression (PS2 + 25−49%) had a confirmed response[64]. Notably, the indication for MIRV monotherapy in PROC is in tumors with FOLR1 expression ≥75%[16]. However, among the 137 tumors we pre-screened for FOLR1 expression, only 14.6% exhibited ≥75% FOLR1 expression. While

there was a trend toward higher response rates in patients with tumors with higher FOLR1 expression, one confirmed, and 1 unconfirmed partial response were observed in patients with tumors with FOLR1 PS2 + 50−74%. Taken together, these data suggest that a FOLR1 cutoff of 50%, or potentially even lower, may be sufficient for clinical activity and also allow expanded access to MIRV-containing therapies in patients with EC. However, this will need to be explored and confirmed in larger cohorts. In addition, FOLR1 expression for the current study was determined from archival tumor tissue. While FOLR1 expression is preserved between original diagnosis and recurrence biopsies in PROC despite intervening lines of therapy[22], this has not yet been established in EC and would require fresh tumor biopsies for comparison to archival tissue in future studies.

The small sample size and early closure of this trial are limitations of the study. Given this, the exploratory subgroup analyses are viewed as descriptive patterns that can inform future hypotheses but will require evaluation in larger cohorts. Similarly, the lack of adjustment for multiple comparisons in the biomarker score analyses makes the composite B score also hypothesis-generating and requires external validation. However, given the broad interest in ADC and ICI combinations in many cancer types, these analyses may serve as a foundation for developing similar composite methods to score the overall favorability of the TIME for ADC and ICI response. Furthermore, this study proposes that expression of the ADC target (in this case, FOLR1) on immune cells (in this case, on tumor-associated macrophages) may be an additional determinant of response to ADC/ICI combinations beyond expression of the ADC target on cancer cells, possibly reflecting on-target, off-tumor immune editing conferred by the ADC that can facilitate response to ICI.

In conclusion, the combination of a FOLR1-targeting ADC, MIRV, and anti-PD-1 inhibition with pembrolizumab demonstrated promising clinical activity and a tolerable safety profile in patients with recurrent FOLR1-expressing, pMMR, serous EC. A composite biomarker score incorporating tumor and macrophage FOLR1 expression and markers of immune activation in the TME was associated with clinical response to MIRV and pembrolizumab in this small cohort and warrants additional investigation in larger cohorts. Overall, our results lend support to ADC and ICI combinatorial therapy and underscore the importance of novel biomarkers of response to these combinations.

## Methods

### Study design and treatment

This was a single-arm, two-stage phase 2 study conducted at 3 centers: the Dana-Farber Cancer Institute (Boston, MA), UMass Memorial Medical Center (Worcester, MA) and Northwell Health Center Institute (New Hyde Park, NY). The study was reviewed and approved by the Dana-Farber and Harvard Cancer Center Institutional Review Board in 2018 and was conducted in accordance with the study protocol, Good Clinical Practice, and the Declaration of Helsinki. The study was preregistered on February 11, 2019, and registered at ClinicalTrials.gov (NCT03835819). All participants provided written informed consent separately to prescreening for FOLR1 expression in archival tumor tissue and then for treatment on study. The first and last patients were enrolled on 2/14/2020 and 5/1/2024, respectively.

Patients with histologically confirmed recurrent or persistent MSS/pMMR, serous (or predominantly serous mixed histology) EC who had received at least one and not more than four prior lines of systemic therapy were eligible for pre-screening to determine FOLR1 expression. Initially, patients were ineligible if they had received any prior immunotherapy, including anti-PD-1 or anti-PD-L1. However, to reflect the changing treatment landscape for EC patients, an amendment was introduced on January 6, 2022 (after enrollment of the first 5 patients) to broaden eligibility criteria to allow prior anti-PD-1/PD-L1

therapy, so long as this was not discontinued due to therapy-related toxicities. Additional key eligibility criteria included adults ages 18 years or older with measurable disease, an Eastern Cooperative Oncology Group (ECOG) performance status of 0 or 1 and adequate organ function as specified in the protocol. Additional exclusion criteria included known active CNS disease, active or chronic corneal disorders, active autoimmune disease, history of pneumonitis or inflammatory lung disease, and history of allergy to monoclonal antibody therapy. For participants who signed pre-screening consent, FRα expression was assessed by central immunohistochemistry (IHC) testing with the anti-FOLR1 2.1 antibody (Ventana Medical Systems) using the PS2+ scoring methodology of quantifying the percentage of tumor cells with staining at ≥2+ intensity. Eligibility for the main study required FOLR1 PS2+ staining of ≥50% of tumor cells. The full inclusion and exclusion criteria are provided in the study protocol in the Supplementary Information.

All participants in the study received pembrolizumab at a fixed dose of 200 mg intravenously and MIRV at a starting dose of 6 mg/kg adjusted ideal body weight intravenously every 3 weeks. Participants were required to undergo a baseline ophthalmic exam during screening and to have repeat ophthalmic exam every 2 cycles and at the end of treatment or at the 30-day follow-up visit. As prophylaxis for ocular symptoms, participants were required to use preservative-free lubricating artificial tears daily throughout treatment and corticosteroid eye drops starting the day before each dose and continuing through day 8 of each cycle as detailed in the study protocol. The combination treatment was continued until progression or adverse event up to 35 cycles; beyond 35 cycles, MIRV could be continued until progression or adverse event. If either study drug was discontinued, patients were permitted to continue the other agent as monotherapy as tolerated. Response to treatment was assessed every 6 weeks according to Response Evaluation Criteria in Solid Tumors (RECIST) version 1.1. Participants who experienced disease progression but remained clinically stable were given the option to continue treatment, with repeat response assessment after 4 weeks to confirm progression, at the discretion of the investigator. Participants who stopped one drug due to toxicity were permitted to continue receiving the other drug as a single agent. Subsequent therapies were not recorded as part of the conduct of the trial.

## Study endpoints and assessments

The primary objective of the study was to assess the clinical activity of MIRV in combination with pembrolizumab in patients with recurrent or persistent MSS/pMMR and FRα-positive EC by evaluating the co-primary endpoints of ORR (defined as confirmed complete and partial responses according to RECIST v1.1), and rate of progression-free survival at 6 months (PFS6), defined as having disease assessment scans for at least 6 months or longer showing no progression per RECIST v1.1. Patients with less than 6 months of follow-up for PFS were treated as non-PFS6. The secondary objectives were to assess the safety and tolerability of MIRV combined with pembrolizumab, as well as additional measures of clinical activity, including PFS, OS, duration of response (DOR), immune-related ORR (irORR), immune-related PFS (irPFS). Although irORR and irPFS were included as secondary endpoints, analysis of ORR and PFS were only reported as per the standard, non immune-related RECIST 1.1, similar to reports of other immunotherapy combination studies in EC, such as KEYNOTE-775[65], nivolumab/cabozantinib[66], avelumab/axitinib[67] as well as the recently reported pembrolizumab and mirvetuximab study in ovarian cancer[68]. Exploratory objectives were to correlate clinical responses with tumor-infiltrating lymphocytes, other tumor-infiltrating and circulating immune cell populations, expression of immune checkpoint proteins, and tumor cell somatic alterations, and to assess the activity of the combination in patients with and without prior exposure to ICI therapy.

The analysis population for both safety and efficacy included all patients who received at least one dose of protocol therapy (modified intent-to-treat). Efficacy endpoints, including ORR, PFS6, PFS, DOR, irORR, and irPFS, were based on investigator-assessed tumor response according to RECIST 1.1 criteria. Radiologic tumor assessments were performed using computerized tomography (CT) scans or magnetic resonance imaging at baseline and then every 6 weeks (2 cycles) while receiving study therapy until time of progression. Participants who discontinued study drugs were followed for 3 years by phone after removal from study or until death. Safety and tolerability of the combination were assessed by recording adverse events using the Adverse Event Severity scale based on the NCI Common Terminology Criteria for Adverse Events version 5.0.

## FOLR1 assessment

FOLR1 status was assessed centrally at Ventana Medical Systems by immunohistochemistry on archival tumor tissue using the anti-FOLR1 clone FOLR1-2.1 antibody (Roche Diagnostics). Additional exploratory analyses were performed on archival tumor tissue to assess the immune composition of tumors and correlate with response to MIRV and pembrolizumab.

## Molecular subtyping and targeted NGS

*POLE* mutation status, TMB, and the presence of additional somatic alterations were determined from archival, formalin-fixed tissues, using targeted-panel next-generation sequencing (Oncopanel; Dana Farber Cancer Institute, Boston, MA) performed at Dana-Farber Cancer Institute[69,70]. MMR status was determined by IHC, by polymerase chain reaction (PCR), and genomically by assessment of the mutational signature as determined by Oncopanel.

## Multiplexed immunofluorescence

Multiplexed immunofluorescence staining was performed using a BOND RX fully automated stainer (Leica Biosystems) and counterstained with Spectral DAPI (4',6-diamidino-2-phenylindole; Akoya Biosciences) as previously described[71]. The target antigens, antibody clones, and dilutions for the two multiplex panels are listed in Supplementary Tables 1 and 2. The antibody clones and staining conditions used for the multiplex panels are validated using chromogenic IHC staining as the baseline[72]. Sample slides are stained on the autostainer in batches along with positive control slides that verify the consistency of staining within and between batches. After staining, slides were cover slipped and loaded into a PhenoImager HT 1.0 (Akoya Biosciences) for whole slide scanning at a 20× magnification. For each sample, three to eight representative regions of interest corresponding to a 20× field of view were selected under pathologist (SJR) supervision using Phenochart 1.0 software (Akoya Biosciences). Following image acquisition, each field of view was spectrally unmixed and analyzed with inForm 2.6 Image Analysis Software (Akoya Biosciences). All nucleated cells within a field of view were segmented via nuclear DAPI staining. A supervised machine-learning algorithm within the software was utilized to assign each cell a phenotype according to biomarker expression within its nuclear and membrane compartments. Positive cell identification was guided by a pathologist (SJR) and visually confirmed. Cell densities (number of positive cells per mm²) and percent positive populations were calculated for each marker.

## Biomarker score calculation

Each tumor was categorized as high or low for each component of the composite biomarker score. For FOLR1 IHC, tumors with 50–74% PS2+ staining were considered low and 75–100% were considered high. For CD8+ cell density, a cutoff of 200 positive cells/mm² was used for low vs high tumors. For PD-L1+ and CD163 + FOLR1+ cell densities, cutoffs of 25 positive cells/mm² and 85 positive cells/mm² were used, respectively.

## Statistical analysis

Statistical considerations were developed for two primary objectives of objective response rate (ORR) and rate of PFS6, using a two-stage design that allowed for early stopping for futility. A two-stage test was constructed using the method of Sill, Rubinstein, Litwin, and Yothers with the goal of stopping early for futility to limit patient exposure to an inactive regimen while restricting the probabilities of type I and type II errors to approximately 10% and 15%, respectively[73]. A true ORR of 5% or less and a rate of PFS6 of 10% or less would not be of clinical interest ($H_0$: OR ≤ 5% AND PFS6 ≤ 10%), whereas an improvement to a 20% ORR or 30% PFS6 rate would warrant further investigation of mirvetuximab and pembrolizumab. In the first stage, 16 patients would be enrolled and accrual would pause. If we observed 2 or more objective responses or 2 or more patients were progression-free at 6 months, accrual would continue to the second stage, where an additional 19 patients would be enrolled. With this design and assuming independence of the endpoints, the probability of early termination for futility under the null is 0.417. The study type I error is 9.8%, and we would have 88.9% power to reject the null when the true ORR is 20% and the true PFS6 rate is 10%, and 87.3% power to reject the null when the true ORR is 5% and the PFS6 rate is 30%. Note that since the study was non-blinded, if we met the criteria to continue to enrollment at any point during the first stage, we could then open enrollment to the target accrual of 35 patients in total. Overall, if ≥4 treated patients with an objective response OR ≥ 8 patients who are progression-free at 6 months were observed, mirvetuximab and pembrolizumab would be considered worthy of further study.

The study met the continuation criteria of 2 objective responses in Stage 1 after the first 5 patients were enrolled. Once these responses were confirmed and the statistical team was consulted, the study opened to full enrollment of 35 patients. After consideration of the rapidly changing treatment landscape in EC and slow study enrollment, completing the trial was deemed unfeasible, and the trial was closed after enrollment of 18 patients.

ORR was defined as the rate of achieving either complete or partial response by RECIST 1.1 among patients who initiated protocol therapy. PFS6 was defined as a binary outcome indicating whether or not a patient was deemed progression-free for at least 6 months. Kaplan–Meier estimates of the progression-free survival function were also estimated, which incorporated censored outcomes in the analysis. 95% confidence intervals (CI) are reported for events rates at landmark times and the median using Greenwood's formula. The statistical significance of biomarker associations with ORR was evaluated using Fisher's exact test, with the missing information excluded from the significance testing. For each type of immune profile data and each region, comparisons were made between clinical groups of interest (defined by objective response) using Wilcoxon rank-sum tests. No adjustments for multiple comparisons were performed. All biomarker analyses are considered exploratory.

## Reporting summary

Further information on research design is available in the Nature Portfolio Reporting Summary linked to this article.

## Data availability

The study protocol and the statistical analysis plan are available in the Supplementary Information file. Additional individual de-identified participant data can be shared upon request. The remaining data are available within the Article or Source Data file. Source data are provided with this paper.

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

## Acknowledgements

We thank all the patients and their families for their participation in this trial. This investigator-initiated study (IND holder P.A.K.) was funded by Merck and AbbVie, which also provided pembrolizumab and mirvetuximab soravtansine, respectively. We would also like to acknowledge support from the Friends of Dana Farber (R.L.P.), the Breast Cancer Research Foundation, and The Lewin Fund to Fight Women's Cancers.

## Author contributions

R.L.P. and J.V. conceived the study; R.L.P. coordinated, analyzed, and interpreted all the data and wrote the manuscript. P.A.K. supervised and was the IND holder of the study. Y.Z., N.T., and N.X. performed the statistical analysis and were the lead statisticians of the study; R.L.P., E.K.L., C.K., S.C., A.A.W., J.F.L, E.H.S., S.Z., V.J., J.V., U.A.M., and P.A.K. provided clinical data and contributed to the analysis and interpretation of the data; N.E., K.L.P., and S.J.R. performed the immunoprofiling data and analysis; M.H., M.P., and H.S. were Clinical Research Project Managers for the study and coordinated the processing and distribution of the clinical trial samples. All authors contributed to the writing and editing of the manuscript.

## Competing interests

R.L.P. declares consulting and/or advisory board participation for Gilead, unrelated to this work; Payment for educational events: OncLive/MJH Life Sciences. E.K.L. declares advisory board participation: Oncusp Therapeutics and Genmab, unrelated to this work, and institutional research support from: Repare Therapeutics, KSQ Therapeutics, Genmab, GSK, Merck, OnCusp Therapeutics, and NiKang Therapeutics. J.F.L. declares consulting and/or advisory board participation for AbbVie, AstraZeneca, Bristol-Myers Squibb, Clovis Oncology, Daiichi Sankyo, Eisai, Genentech/Roche, Genmab, GlaxoSmithKline, LoxoLilly, Merck, SystImmune, Regeneron Therapeutics, Revolution Medicine, and Zentalis Pharmaceuticals, unrelated to this work. S.J.R. declares research support from Coherus Pharmaceuticals, Delcath Pharmaceuticals, and Bristol Myers Squibb. S.R. is a member of the Scientific Advisory Board of Immunitas Therapeutics. J.V. declares current employment at GSK, unrelated to this work. U.A.M. declares participation in scientific advisory boards: NextCure, Abbvie, Immunogen, Profound Bio, Eisai, the Ovarian Cancer Research Alliance, Tango Therapeutics, Novartis, GSK, Daiichi Sankyo, DayOne Bio, and Whitehawk Therapeutics; UAM also reports participation in a data safety-monitoring board: Mural Oncology, Macrogenics, Daiichi Sankyo, Astrazseneca and Symphogen, all unrelated to this work. P.A.K. declares consulting and/or advisory board participation for AstraZeneca, Bayer, GSK, Merck, Pfizer, BMS, Repare, IMV, Artios, Kadmon, Cardiff, Immunogen, EMD Serono, Scorpion, Schrodinger, Nimbus, Mural Oncology, unrelated to this work. Y.Z., N.E., M.H., M.P., C.K., S.C., A.A.W., E.H.S., H.S., N.X., K.L.P., N.T., S.Z., V.J. declare no competing interests.

## Additional information

Rebecca L. Porter ®[1] ✉, Yinglu Zhou ®[1], Nebiyat Eskndir[1], Martin Hayes[1], Madeline Polak ®[1], Elizabeth K. Lee ®[1], Carolyn Krasner[1], Susana Campos[1], Alexi A. Wright[1], Joyce F. Liu[1], Elizabeth H. Stover ®[1], Hannah Sawyer[1], Niya Xiong[1], Kathleen L. Pfaff[1], Scott J. Rodig ®[2], Nabihah Tayob ®[1], Susan Zweizig[3], Veena John[4], Jennifer Veneris[1], Ursula A. Matulonis[1] & Panagiotis A. Konstantinopoulos ®[1]

[1]Dana-Farber Cancer Institute, Boston, MA, USA. [2]Brigham & Women's Hospital, Boston, MA, USA. [3]University of Massachusetts Memorial Medical Center, Worcester, MA, USA. [4]Northwell Health Cancer Institute, New Hyde Park, NY, USA. ✉e-mail: rebecca_porter@dfci.harvard.edu

