## [Transparent Peer Review file · Nature Communications]

Mirvetuximab Soravtansine plus Pembrolizumab in Recurrent Folate Receptor Alpha-Positive Uterine Serous Carcinoma: a phase II trial

Corresponding Author: Dr Rebecca Porter

Version 0:

Reviewer comments:

Reviewer #2

(Remarks to the Author)

The authors have edited the manuscript appropriately based on the comments from the reviewers. The reviews were overall positive to begin with unfortunately with the smaller numbers the enthusiasm is a little more muted. With that being said re-reviewing the manuscript, the responses and the global story presented with the therapeutic angle and translational angle it's an excellent manuscript.

Reviewer #3

(Remarks to the Author)

I would like to thank the editorial office for the opportunity to review the revised manuscript submitted by Porter et al. The authors describe a single arm phase 2 trial examining the combination of mirvetuximab + pembrolizumab in patients with recurrent serous endometrial cancer.

I have a few comments/considerations for the authors:

- In the methods section, please clarify if FOLR1 expression levels were adjudicated locally, by the institutional pathologist? Also, was FOLR1 assessment done using the current approved platform in patients with epithelial ovarian cancer. I anticipate this may be the case, but would include this results paragraph detailing FOLR1 expression in the cohort.
- In lines 104-105 it stated that 20 tumors were found to have FOLR1 expression $\geq 75\%$ of tumor cells. In the subsequent paragraph, lines 115, it states that 50% of patients had FOLR1 $\geq 75\%$. The population references should be clarified to avoid confusion.
- Was the median PFS of 2.73 months equivalent to the imaging assessment interval on trial?
- If the authors report on OS (median OS in the overall cohort was 12.5 months), it would be beneficial to understand subsequent therapies. With a small cohort, this should be available and will help contextualize the reported median OS value. As an example, did the patients with HER2 2+/3+ treated with T-DXd?
- The authors report that the median PFS in the ICI-exposed cohort was 1.2 months. Were these patients scanned earlier than anticipated per protocol for clinical symptoms of disease progression?
- There is a typo on line 182, it reads "profiling of on..". I believe that the word "on" needs to be removed.
- Comparing FOLR1 by IF and IHC in non-contiguous tumor samples may limit the interpretation of the data substantially. Also, were these assessments on the same tissue block, or from alternate sites of metastatic disease?
- Although interesting, the biomarker analysis needs to be controlled for multiple testing. Did the authors divide the significance threshold by the number of tests being performed, or the number of potential biomarker combinations explored? Alternatively, a Hochberg correction can be considered. It is unclear to me if this was done? I am concerned that too much weight is being placed on this exploratory analysis in the discussion.

Reviewer #4

Summary of study design and analyses This is a single-arm, two-stage phase 2 trial evaluating mirvetuximab soravtansine (MIRV) plus pembrolizumab in 18 patients with recurrent/persistent pMMR serous endometrial cancer (FOLR1 expression $\geq 50\%$). The study utilized a dual-endpoint two-stage design (Sill et al.) with co-primary endpoints of Objective Response Rate (ORR) and 6-month Progression-Free Survival (PFS6). The trial was designed for 35 patients but closed early due to feasibility issues rather than futility.

Among the 18 treated patients, the confirmed ORR was 28% (95% CI 10–53%), and the Kaplan–Meier estimated PFS6 was 24.4%. All confirmed responses occurred in ICI-naïve patients. Safety was consistent with known profiles. Exploratory analyses suggested potential associations between response and CCNE1 amplification or a composite biomarker ("B score").

Major comments

1. **Early closure and two-stage design implications** The trial was designed for 35 patients with specific operating characteristics (Type I $\approx 10\%$, Type II $\approx 15\%$) but closed after 18 patients. While the operational decision is reasonable, it invalidates the originally specified error control and decision boundaries.

Recommendation: Explicitly state in the Methods/Results that while the trial met the stage 1 continuation rule (≥ 2 responses), the full two-stage design was not completed. Consequently, the reported results should be framed as preliminary and hypothesis-generating rather than as meeting formal phase 2 success criteria.

2. **Co-primary endpoints and success criteria** The use of a dual-endpoint design is a strength. However, the specific success criteria for the co-primary endpoints need better visibility.

Recommendation: Explicitly state the exact success criterion (e.g., " ≥ 4 responses OR ≥ 8 patients progression-free at 6 months") to clarify how multiplicity was handled. Additionally, briefly summarize the actual operating characteristics under plausible scenarios rather than citing approximate error rates.

3. **Definition of PFS6 (Design Count vs. KM Estimate)** The design utilizes a simple count of patients progression-free at 6 months, yet the Results report a Kaplan–Meier estimate (24.4%).

Recommendation: Please distinguish between the design quantity (binary count in the $n=35$ design) and the KM-estimated PFS6. Clarify if all 18 patients had sufficient follow-up for the binary endpoint or how censoring was handled regarding the stopping decision.

4. **Population heterogeneity (Protocol Amendment)** The amendment allowing prior ICI exposure introduces heterogeneity. Prior ICI exposure was associated with shorter PFS (HR = 0.2), complicating efficacy interpretation.

Recommendation: Highlight that the original design assumptions implicitly assumed a homogeneous population. The subgroup findings (ICI-naïve vs. exposed) should be labeled as exploratory and unadjusted for confounders given the sample size.

5. **Biomarker analyses and overfitting** The biomarker work is a strength but is statistically underpowered. The composite "B score" was derived and evaluated in the same small cohort, carrying a high risk of overfitting.

Recommendation: Explicitly state that no adjustment for multiple comparisons was performed. Clearly label the B score as a hypothesis-generating composite requiring external validation. I suggest simplifying the narrative to focus on biological signals (e.g., CCNE1, TIME) rather than specific p-values or cut-points.

6. **Estimation and uncertainty** Given the sample size ($n=18$), the precision of estimates is low.

Recommendation: Emphasize the wide confidence intervals for ORR (10–53%) in the Discussion. Comparisons by prior ICI or FOLR1 status should be framed as descriptive patterns rather than definitive subgroup effects.

Minor comments / suggestions

Analysis Population: Explicitly define the analysis population (e.g., mITT). Confirm that all treated patients, including early withdrawals, were included in the ORR denominator.

Subgroup Visualization: A forest plot or table summarizing ORR/PFS by ICI status and FOLR1 expression levels (50–74% vs. $\geq 75\%$) would clarify the uncertainty around these subgroup effects.

Feasibility: You note that only $\sim 15\%$ of prescreened tumors met the $\geq 75\%$ FOLR1 cutoff. A brief comment on the implications of this for future trial feasibility and screening volume would be valuable.

Overall assessment This trial addresses an important question using a sensible design. The primary limitations—early closure ($n=18$ vs. 35), small sample size, and lack of multiplicity control in exploratory analyses—do not invalidate the study

but require careful framing. With the clarifications suggested above, the statistical reporting will be appropriate for a signal-seeking phase 2 study.

Version 1:

Reviewer comments:

Reviewer #3

(Remarks to the Author)

The revised manuscript was reviewed. The authors/study team have thoughtfully and comprehensively addressed all concerns. I find the manuscript suitable for publication in the current format.

Reviewer #4

(Remarks to the Author)

The authors carefully addressed all of my questions and concerns. I have no further major concerns.

One minor suggestion relates to the power and Type I error calculation under endpoint dependence. The authors state: 'When the endpoints are assumed to be dependent, the study power is 84.6% and 86.2%, respectively. The type I error is 9.8% assuming independence of endpoints and 0.088 assuming dependence of endpoints.'

Statistical Clarification: Since dependence is determined by the correlation (ρ), the dependence scenario is not unique. Could the authors please specify the assumed correlation coefficient (ρ) between the co-primary endpoints that resulted in the stated power (86.2%) and Type I error (0.088)?

Alternatively, to simplify the discussion, the authors may consider removing the dependence scenario and focusing solely on the operating characteristics calculated under the independence assumption.

Phase II Study of Mirvetuximab Soravtansine plus Pembrolizumab in Recurrent Folate Receptor Alpha-Positive Uterine Serous Carcinoma

Response to Reviewer Comments

We thank the reviewers for the thorough and considerate review and for including helpful comments. We have carefully considered the reviewers' remarks and have responded point-by-point below (in bold) and revised the manuscript accordingly.

Reviewer #2 (Remarks to the Author):

The authors have edited the manuscript appropriately based on the comments from the reviewers. The reviews were overall positive to begin with unfortunately with the smaller numbers the enthusiasm is a little more muted. With that being said re-reviewing the manuscript, the responses and the global story presented with the therapeutic angle and translational angle it's an excellent manuscript.

We thank Reviewer #2 for re-reviewing our manuscript and for their overall positive assessment.

Reviewer #3 (Remarks to the Author):

I would like to thank the editorial office for the opportunity to review the revised manuscript submitted by Porter et al. The authors describe a single arm phase 2 trial examining the combination of mirvetuximab + pembrolizumab in patients with recurrent serous endometrial cancer.

I have a few comments/considerations for the authors:

- In the methods section, please clarify if FOLR1 expression levels were adjudicated locally, by the institutional pathologist? Also, was FOLR1 assessment done using the current approved platform in patients with epithelial ovarian cancer. I anticipate this may be the case, but would include this results paragraph detailing FOLR1 expression in the cohort.

We thank the reviewer for pointing out the need for further clarification on FOLR1 assessment. It is correct that the current approved Ventana platform was used in this study, and this was performed centrally at Ventana. This is stated in the Results section under "FOLR1 assessment" on line 492: "FOLR1 status was assessed centrally at Ventana Medical Systems by immunohistochemistry on archival tumor tissue using the anti-FOLR1 clone FOLR1-2.1 antibody (Roche Diagnostics)."

We have also added this clarification to the Results section as suggested on line 100: "In the current study, a total of 144 eligible participants signed consent for pre-screening of archival tumor tissue for FOLR1 expression by central IHC using the anti-FOLR1 2.1 antibody developed by Ventana Medical Systems and following the PS2 scoring methodology of quantifying the percentage of tumor cells staining at $\geq 2+$ intensity."

- In lines 104-105 it stated that 20 tumors were found to have FOLR1 expression $\geq 75\%$ of tumor cells. In the subsequent paragraph, lines 115, it states that 50% of patients had FOLR1 $\geq 75\%$. The population references should be clarified to avoid confusion.

We have specified in line 117 (previously line 115) that the 50% of patients with FOLR1 $\geq 75\%$ was in the enrolled population, whereas the 20 patients (14.6%) with FOLR1 $\geq 75\%$ was in the prescreening population.

- Was the median PFS of 2.73 months equivalent to the imaging assessment interval on trial?

Imaging assessment was performed every 6 weeks (+/- 1 week) on study. The median PFS of 2.73 months (i.e., $2.73 * 30.4 = 82.99$ days = 11.86 weeks) corresponds to roughly the second imaging assessment.

- If the authors report on OS (median OS in the overall cohort was 12.5 months), it would be beneficial to understand subsequent therapies. With a small cohort, this should be available and will help contextualize the reported median OS value. As an example, did the patients with HER2 2+/3+ treated with T-DXd?

We thank the reviewer for this question. Subsequent therapies were not officially recorded as part of the conduct of the trial. However, of the 19 patients in the study, subsequent treatment information was available for 11 patients. 5 patients received one more regimen, 2 had received two regimens and 5 patients received 3 or more subsequent regimens (maximum 6). Subsequent regimens included chemotherapy (such as PLD, carbo/PLD, gemcitabine), immunotherapy/antiangiogenic therapy (pembro/lenvatinib), targeted therapies (Enhertu, PARPi, PI3K targeted therapies) and clinical trial participation. We have added in the manuscript Methods that subsequent therapies was not recorded as part of the study.

- The authors report that the median PFS in the ICI-exposed cohort was 1.2 months. Were these patients scanned earlier than anticipated per protocol for clinical symptoms of disease progression?

The median PFS in the ICI-exposed cohort of 1.2 months ($1.2 * 30.4 = 36.48$ days = 5.21 weeks) corresponds to the first imaging assessment (6 weeks +/- 1 week) as specified in the protocol; no patients were scanned earlier than anticipated.

- There is a typo on line 182, it reads "profiling of on..". I believe that the word "on" needs to be removed.

We thank the reviewer for pointing this out; the manuscript file has been corrected.

- Comparing FOLR1 by IF and IHC in non-contiguous tumor samples may limit the interpretation of the data substantially. Also, were these assessments on the same tissue block, or from alternate sites of metastatic disease?

We thank the reviewer for recognizing this potential limitation of the biomarker analyses. We agree that contiguous tumor samples would be ideal for assessing FOLR1 IHC and IF. However, contiguous slides were not available in all samples and therefore the assays were performed on the available tissue. In general, as shown in the representative images in Supplemental Fig 4, similar tumor sections were selected for IHC and IF staining and we observed good concordance between the two assays. We have acknowledged this issue in the manuscript (line 242) as well.

- Although interesting, the biomarker analysis needs to be controlled for multiple testing. Did the authors divide the significance threshold by the number of tests being performed, or the number of potential biomarker combinations explored? Alternatively, a Hochberg correction can be considered. It is unclear to me if this was done? I am concerned that too much weight is being placed on this exploratory analysis in the discussion.

Please see our response to Reviewer #4 comment 5 below.

Reviewer #4 (Remarks to the Author):

We thank the statistical Reviewer for the opportunity to improve our manuscript with their very helpful comments and recommendations. We have incorporated and responded to each in a point-by-point response below.

Statistical / Design Review

Summary of study design and analyses This is a single-arm, two-stage phase 2 trial evaluating mirvetuximab

soravtansine (MIRV) plus pembrolizumab in 18 patients with recurrent/persistent pMMR serous endometrial cancer (FOLR1 expression $\geq 50\%$). The study utilized a dual-endpoint two-stage design (Sill et al.) with co-primary endpoints of Objective Response Rate (ORR) and 6-month Progression-Free Survival (PFS6). The trial was designed for 35 patients but closed early due to feasibility issues rather than futility.

Among the 18 treated patients, the confirmed ORR was 28% (95% CI 10–53%), and the Kaplan–Meier estimated PFS6 was 24.4%. All confirmed responses occurred in ICI-naïve patients. Safety was consistent with known profiles. Exploratory analyses suggested potential associations between response and CCNE1 amplification or a composite biomarker ("B score").

Major comments

1. Early closure and two-stage design implications The trial was designed for 35 patients with specific operating characteristics (Type I $\approx 10\%$, Type II $\approx 15\%$) but closed after 18 patients. While the operational decision is reasonable, it invalidates the originally specified error control and decision boundaries.

Recommendation: Explicitly state in the Methods/Results that while the trial met the stage 1 continuation rule (≥ 2 responses), the full two-stage design was not completed. Consequently, the reported results should be framed as preliminary and hypothesis-generating rather than as meeting formal phase 2 success criteria.

In the Statistical Analysis section on line 555, we state:

“The study met the continuation criteria of 2 objective responses in Stage 1 after the first 5 patients were enrolled. Once these responses were confirmed and the statistical team was consulted, the study opened to full enrollment of 35 patients.”

In addition, we have revised the following text in the results section starting on line 123:

“Among the first 5 patients enrolled, we observed 2 confirmed objective responses meeting the criteria to continue enrollment in the two-stage design. Among all 18 patients, the confirmed ORR was 28% (95% CI: 10-53%) including 1 confirmed complete response, 4 confirmed partial responses (Figure 2A). While the study meets the required number of responses for ORR (≥ 4 responses) specified by the two-stage design, and the 95% confidence interval excludes the null ORR of 5%, we did not complete enrollment of the trial to the planned sample size of 35 patients and hence these results should be considered preliminary.”

2. Co-primary endpoints and success criteria. The use of a dual-endpoint design is a strength. However, the specific success criteria for the co-primary endpoints need better visibility.

Recommendation: Explicitly state the exact success criterion (e.g., " ≥ 4 responses OR ≥ 8 patients progression-free at 6 months") to clarify how multiplicity was handled. Additionally, briefly summarize the actual operating characteristics under plausible scenarios rather than citing approximate error rates.

The success criterion has been clearly stated on line 552 of the manuscript:

“Overall, if ≥ 4 treated patients with an objective response OR ≥ 8 patients who are progression-free at 6 months were observed, mirvetuximab and pembrolizumab would be considered worthy of further study.”

We have also added the following text on line 543 to describe the full detailed operating characteristics of the design.

“With this design, the probability of early termination for futility under the null assuming independence of the endpoints is 0.417 and 0.502 assuming dependence of the endpoints. The study would have 88.9% power to reject the null when the true ORR is 20% and the true PFS6 rate is 10%, and 87.3% power to reject the null when the true ORR is 5% and the PFS6 rate is 30% assuming independence of the endpoints. When the endpoints are assumed to be dependent, the study power is 84.6% and 86.2%,

respectively. The type I error is 9.8% assuming independence of endpoints and 0.088 assuming dependence of endpoints.

3. Definition of PFS6 (Design Count vs. KM Estimate) The design utilizes a simple count of patients progression-free at 6 months, yet the Results report a Kaplan–Meier estimate (24.4%).

Recommendation: Please distinguish between the design quantity (binary count in the n=35 design) and the KM-estimated PFS6. Clarify if all 18 patients had sufficient follow-up for the binary endpoint or how censoring was handled regarding the stopping decision.

We have added the definition of PFS6 to manuscript on line 470:

“(PFS6) defined as having disease assessment scans for at least 6 months or longer showing no progression per RECIST v1.1. Patients with less than 6 months of follow-up for PFS were treated as non-PFS6.”

We have now separated the binary PFS6 estimate from the Kaplan-Meier method estimate more clearly in the Results section on line 134:

“There were 4 patients alive and progression-free at 6 months resulting in a PFS6 rate of 22% (95% CI: 6-48%).”

“The observed median PFS (mPFS) was 2.73 months (95% CI: 1.2-4.5 months) (Supplemental Figure 2) and the PFS at 6 months was 24.4% (95% CI 7.7-46.1%) by the Kaplan-Meier method.”

4. Population heterogeneity (Protocol Amendment) The amendment allowing prior ICI exposure introduces heterogeneity. Prior ICI exposure was associated with shorter PFS (HR = 0.2), complicating efficacy interpretation.

Recommendation: Highlight that the original design assumptions implicitly assumed a homogeneous population. The subgroup findings (ICI-naïve vs. exposed) should be labeled as exploratory and unadjusted for confounders given the sample size.

We have modified this section of the results on line 147 and added the following text prior to reporting these results:

“The initial design of the trial enrolled ICI naïve patients only but a subsequent protocol amendment allowed for enrollment of up to 19 patients with prior ICI targeting the PD-1/PD-L1 pathway. In an exploratory subgroup analysis, we evaluated the efficacy of mirvetuximab and pembrolizumab in patients that were ICI naïve (n=10) versus exposed (n=8).”

5. Biomarker analyses and overfitting The biomarker work is a strength but is statistically underpowered. The composite "B score" was derived and evaluated in the same small cohort, carrying a high risk of overfitting.

Recommendation: Explicitly state that no adjustment for multiple comparisons was performed. Clearly label the B score as a hypothesis-generating composite requiring external validation. I suggest simplifying the narrative to focus on biological signals (e.g., CCNE1, TIME) rather than specific p-values or cut-points.

In the Statistical methods section on line 569 we have added the following text:

“No adjustments for multiple comparisons were performed. All biomarker analyses are considered exploratory.”

In the Discussion section on line 342, we have added the following text when discussing the B score:

“However, *this score was derived and evaluated in this small cohort and should therefore be considered hypothesis-generating. Validation in external cohorts of patients in the future will be required to draw conclusions on the clinical relevance of this B score.*”

We also expanded the summary statement regarding the B score on line 264 to focus more on the TIME rather than specific score, as suggested:

“Median overall survival was 1.28 months (95% CI: 1.18–NR) for patients with B– scores (n=7, 6 events) compared with 6.19 months (95% CI: 2.73–NR) for patients with B+ scores (n=8, 5 events), supporting the potential clinical significance of this *hypothesis-generating biomarker score, or others similarly describing a favorable tumor and immune microenvironment*, in predicting patients who may derive benefit from an ADC combined with ICB.”

6. Estimation and uncertainty Given the sample size (n=18), the precision of estimates is low.

Recommendation: Emphasize the wide confidence intervals for ORR (10–53%) in the Discussion. Comparisons by prior ICI or FOLR1 status should be framed as descriptive patterns rather than definitive subgroup effects.

We have addressed this on line 304 in the discussion of comparisons by prior ICI or FOLR1 status:

“Given the smaller size of this study, associations between clinical activity and FOLR1 expression and/or prior ICI exposure will need to be confirmed in future studies.”

And have additionally added a limitations paragraph at the end of the discussion (line 388) to more explicitly state these limitations:

“The small sample size and early closure of this trial are clear limitations of the study. Given this, the exploratory subgroup analyses are viewed as descriptive patterns that can inform future hypotheses but will require evaluation in larger cohorts. Similarly, the lack of adjustment for multiple comparisons in the biomarker score analyses makes the composite B score also hypothesis-generating and requires external validation. However, given the broad interest in ADC and ICI combinations in many cancer types, these analyses may serve as a foundation for developing similar composite methods to score the overall favorability of the tumor immune microenvironment for ADC and ICI response.”

Minor comments / suggestions

Analysis Population: Explicitly define the analysis population (e.g., mITT). Confirm that all treated patients, including early withdrawals, were included in the ORR denominator.

Yes, we used a modified intent to treat analysis population where all patients that received any protocol therapy were included in the safety and efficacy populations. Of note, one patient was incorrectly assigned a study ID (#18) but did not enroll in the trial and hence they were not included in the analysis population.

Subgroup Visualization: A forest plot or table summarizing ORR/PFS by ICI status and FOLR1 expression levels (50–74% vs. ≥75%) would clarify the uncertainty around these subgroup effects.

Supplementary Tables 2 and 4 summarize the ORR/median PFS by ICI status and FOLR1 expression levels (50–74% vs. ≥75%), respectively. These have been modified to include ORR and associated confidence intervals and HR and associated confidence intervals to be comprehensive per the reviewer’s suggestion.

Feasibility: You note that only ~15% of prescreened tumors met the $\geq 75\%$ FOLR1 cutoff. A brief comment on the implications of this for future trial feasibility and screening volume would be valuable.

The 75% reference in the discussion around FOLR1 expression refers to the cutoff for clinical use of mirvetuximab in ovarian cancer, not in endometrial cancer. We were raising the point that activity was seen in our lower cut-off of 50% which may allow for improved access to the drug in patients with endometrial cancer. To make the reference to ovarian cancer clearer, we have spelled out PROC to state “ovarian cancer”.

Overall assessment This trial addresses an important question using a sensible design. The primary limitations—early closure ($n=18$ vs. 35), small sample size, and lack of multiplicity control in exploratory analyses—do not invalidate the study but require careful framing. With the clarifications suggested above, the statistical reporting will be appropriate for a signal-seeking phase 2 study.

Reviewer #4 (Remarks to the Author):

One minor suggestion relates to the power and Type I error calculation under endpoint dependence. The authors state: 'When the endpoints are assumed to be dependent, the study power is 84.6% and 86.2%, respectively. The type I error is 9.8% assuming independence of endpoints and 0.088 assuming dependence of endpoints.'

Statistical Clarification: Since dependence is determined by the correlation (ρ), the dependence scenario is not unique. Could the authors please specify the assumed correlation coefficient (ρ) between the co-primary endpoints that resulted in the stated power (86.2%) and Type I error (0.088)?

Alternatively, to simplify the discussion, the authors may consider removing the dependence scenario and focusing solely on the operating characteristics calculated under the independence assumption.

We thank the reviewer for this suggestion. We agree that the dependence scenario is not unique and have removed it from the manuscript and only present the operating characteristics calculated under the independence assumption.